

# Spring-neap tidal cycles modulate the strength of the carbon source at the estuary-coast interface

Vlad A. Macovei[1], Louise C.V. Rewrie[1], Rüdiger Röttgers[2], Yoana G. Voynova[1]

[1]Helmholtz Zentrum Hereon, Institute of Carbon Cycles, Department of Coastal Productivity; Max-Planck-Str. 1, Geesthacht
21502, Germany
[2]Helmholtz Zentrum Hereon, Institute of Carbon Cycles, Department of Optical Oceanography; Max-Planck-Str. 1,
Geesthacht 21502, Germany

*Correspondence to*: Vlad A. Macovei (vlad.macovei@hereon.de)



**Abstract.** Estuaries are dynamic environments with large biogeochemical variability modulated by tides, linking land to the coastal ocean. The carbon cycle at this land-sea interface can be better constrained by increasing the frequency of observations and by identifying the influence of tides with respect to the spring-neap variability. Here we use FerryBox measurements from a Ship-of-Opportunity travelling between two large temperate estuaries in the North Sea and find that the spring-neap tidal cycle drives a large percentage of the biogeochemical variability, in particular in inorganic and organic

carbon concentrations at the land-sea interface in the outer estuaries and the adjacent coastal region. Of particular importance to carbon budgeting is the up to 74% increase (up to $43.0 \pm 17.1$ mmol C m$^{-2}$ day$^{-1}$) in the strength of the estuarine carbon source to the atmosphere estimated during spring tide in a macrotidal estuary. We describe the biogeochemical processes occurring during both spring and neap tidal stages, their net effect on the partial pressure of carbon dioxide in seawater, and the ratios of dissolved inorganic to dissolved organic carbon concentrations. Surprisingly, while the two example outer

estuaries in this study differ in the timing of the variability, the metabolic state progression and the observed phytoplankton species distribution, an increase in the strength of the potential carbon source to the atmosphere occurs at both outer estuaries on roughly 14-day cycles, suggesting that this is an underlying characteristic essential for the correct estimation of carbon budgets in tidally-driven estuaries and the nearby coastal regions. Understanding the functioning of estuarine systems and quantifying their effect on coastal seas should improve our current biogeochemical models and therefore future carbon

exchange and budget predictability.

**Short Summary.** A commercial vessel equipped with scientific instruments regularly travelled between two large macro-tidal estuaries. We found that biogeochemical variability in the outer estuaries is driven by the 14-day spring-neap tidal cycle, with strong effects on dissolved inorganic and organic carbon concentrations and distribution. Since this land-sea

interface effect increases the strength of the carbon source to the atmosphere by 74% during spring tide, it should be accounted for in regional models.

**Graphical Abstract.**

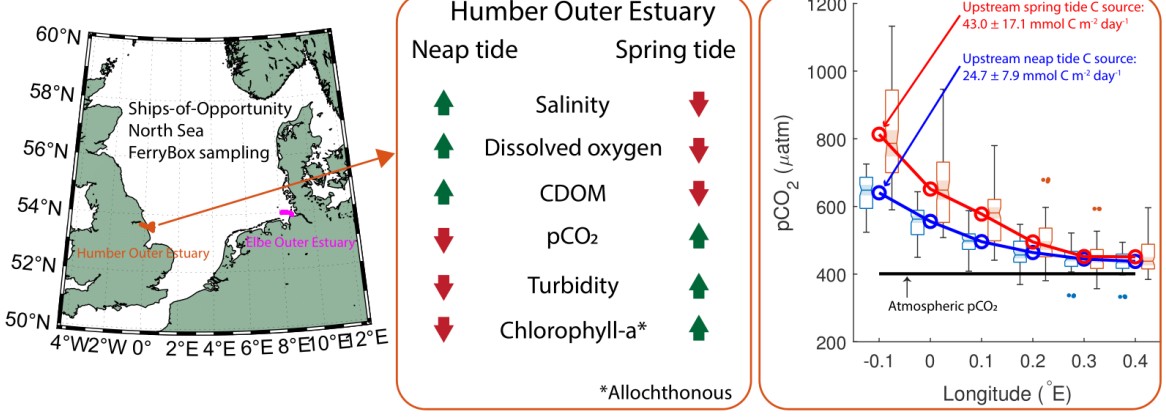



## 1 Introduction

Coastal seas and estuaries are heterogeneous environments characterized by dynamic biogeochemical variability (Bauer et al., 2013), largely driven by river inputs of water, dissolved and suspended matter from land to the coastal ocean (Burson et al., 2016;Frigstad et al., 2020). Estuaries of large rivers are facing biogeochemical variability on regular short-term (day/night biological cycles and diurnal/semi-diurnal tidal cycles) or medium-term scales (synodic month tidal cycles), as

well as irregular variability through flood or drought events affecting the flow rate (Regnier et al., 2013;Shen et al., 2019;Joesoef et al., 2017). They are also affected by anthropogenic activities, which can alter ecosystem functioning, carbon and nutrient cycling, and it can take decades for ecosystems to recover (Rewrie et al., 2023b). Thus, while the variability of estuaries and coastal oceans can be difficult to capture, it is essential to attempt to quantify the key processes driving this variability at the land-ocean interface, including how regional physics can affect biogeochemistry (Canuel and Hardison,

2016;Gattuso et al., 1998).

One particular aspect of the land-ocean continuum (LOC) with large knowledge gaps is the carbon cycle (Legge et al., 2020). Biogeochemical processes, such as primary production, remineralization, carbonate precipitation and dissolution, air-sea and sediment-water exchanges, can all alter the carbonate system over short spatial scales (Cai et al., 2021), and tides add

a further layer of complexity. Generally, estuarine waters are a source of $CO_2$ to the atmosphere (Volta et al., 2016;Riemann et al., 2016;Hudon et al., 2017;Canuel and Hardison, 2016;Chen and Borges, 2009;Borges, 2005), with a global estimate of 0.25 Pg C yr$^{-1}$ (Cai, 2010). In fact, this amount of carbon released to the atmosphere by estuaries could potentially counteract the carbon absorbed by continental shelves (Laruelle et al., 2010). The reasons for high dissolved inorganic carbon concentrations – and implicitly high partial pressure of carbon dioxide ($p$CO$_2$) – in estuaries are the dominance of

heterotrophic processes and direct inputs from rivers, groundwater, or intertidal marshes (Neubauer and Anderson, 2003). Anthropogenic pressures in the water catchment, such as land use change, agriculture and industrial activities, can influence the carbon load and the subsequent biogeochemical transformations further downstream (García-Martín et al., 2021;Wolff et al., 2010). Estuaries and the near-shore coastal area are therefore regions vulnerable to such pressures (Howarth et al., 2011). Tidal variability has been found to influence biogeochemical changes at shelf edges (Sharples et al., 2007;Lucas et al.,

2011). Here we investigate the tidal effect at the estuary-shelf sea interface.

With a good understanding of the underlying processes that govern the variability in the LOC, the carbon biogeochemistry in outer estuaries can be modeled and used to better balance the carbon budgets of shelf seas (Ward et al., 2017;Dai et al., 2022). For example, excluding the effect of estuarine plumes from a computation of the annual CO$_2$ flux in the southern

North Sea increased the sea-to-air flux by 20% (Schiettecatte et al., 2007). Challenges remain – for example, a coupled hydrodynamic-ecosystem model used to investigate the impact of coastal acidification in the North Sea was still not able to reproduce $p$CO$_2$ correctly (Artioli et al., 2012). Moreover, on a global scale, there are many differing coastal systems along



the land-sea interface, such as glaciated fjords, hypersaline estuaries, or coastal lagoons, as well as less-studied regions compared to Europe, the United States and East Asia. Solving the complexity of modeling such heterogeneous land-sea

systems, and particularly those affected by tidal forcing globally is therefore still demanded. Numerous studies found that the best way to capture and assess this heterogeneity is to increase observations, as well as by identifying and correctly characterizing the processes in regions where observations already exist (Regnier et al., 2022;Schiettecatte et al., 2007;Kuliński et al., 2011;Voynova et al., 2015). In an estuarine setting for example, high-frequency observations are particularly important due to the large horizontal gradients present (Cai et al., 2021;Kerimoglu et al., 2018). Furthermore, the

difference in the tidal energy between spring and neap tides influences the location and intensity of mixing processes. A high observational frequency that is able to capture both space and time variability can be achieved with Ship-of-Opportunity (SOO) measurements (Jiang et al., 2019).

In a recent study, Macovei et al. (2022) observed high seawater $pCO_2$ outside the Humber River estuary, as well as

variability that seemed to match the spring-neap tidal cycles. However, the authors found that while the Copernicus Marine Environment Biogeochemical Shelf Sea Model (Butenschön et al., 2016) captured the $pCO_2$ spatial distribution in the central North Sea, in the nearshore, outer estuary region neither the overall $pCO_2$ levels, nor the variability in $pCO_2$ were accurately reproduced. Furthermore, not accounting for the influence of estuaries in coastal regional models, as evidenced by Canuel and Hardison (2016), raises the uncertainty of carbon budget estimations, and can lead to erroneous results. A recent study

showed that rivers perturb the coastal carbon cycle to a larger extent offshore than previously considered (Lacroix et al., 2021), and can influence coastal regions with changes in estuarine discharge (Voynova et al., 2017;Kerimoglu et al., 2020;Garvine and Whitney, 2006). In addition, the land-based inputs and the partitioning between the inorganic and organic carbon forms is needed for regional budget calculations (Kitidis et al., 2019). In this context, this study examines the influence from tidally-driven spring-neap biogeochemical variability in the outer estuary regions, characterizes this

variability with respect to the carbon concentrations and fluxes at the land-sea interface and assesses the largely unaccounted impact of this variability on regional carbon budget assessments.

## 2 Methods

### 2.1 Study area

The North Sea as a whole has been previously characterized as an important carbon sink (Thomas et al., 2005), but recently,

driven by summertime biological activity, a decrease in the buffer capacity and a diminishing of the continental shelf pump (Tsunogai et al., 1999), its carbon uptake capacity has weakened (Lorkowski et al., 2012;Clargo et al., 2015;Bourgeois et al., 2016). The seawater $pCO_2$ in the North Sea was found to increase at a faster rate than the atmospheric one, mainly driven by non-thermal effects in the summer months, shifting the carbon uptake/release boundary northwards (Macovei et al., 2021). In particular, this affects the Central North Sea, which separates the northern region as a dominant sink from the southern





region as an overall source of $CO_2$ to the atmosphere. This creates a fragile balance regarding the direction of the carbon
dioxide flux, which depends on the dominance of thermal or biological forcing (Kitidis et al., 2019;Hartman et al., 2019).
Additionally, several tidally-driven estuaries of major European rivers flow into this shelf sea, where tides have already been
shown to influence primary production (Zhao et al., 2019) and we hypothesize that the regions outside their estuaries exhibit
large $p$CO$_2$ variability that needs to be resolved for correct regional budget assessments. The North Sea is therefore an ideal

location to investigate this balance and variability using long-term and high-frequency observations, as made possible by a
specific SOO route. The MS *Hafnia Seaways* (DFDS Seaways, Copenhagen, Denmark) is a cargo vessel that regularly
transited the North Sea between 2014 and 2018. The ship was equipped with a FerryBox as part of the Coastal Observing
System for Northern and Arctic Seas - COSYNA (Baschek et al., 2017). The two main ports of the MS *Hafnia Seaways*
route during this time were Immingham, UK and Cuxhaven, Germany. These ports are located within the outer estuaries of

the Humber and Elbe Rivers respectively. Therefore, the observations from this ship allowed us to investigate the
biogeochemical variability in two estuary-influenced near-shore areas of the same shelf sea, but with different characteristics
and catchment regions (Fig. 1). While the data in the Central North Sea are available, and have even been used for exploring
the carbon dynamics in the region (Macovei et al., 2021), this work is focused on the land-sea interface, and in particular the
two temperate estuaries and their adjacent coastal regions specified in Fig. 1c and 1d.

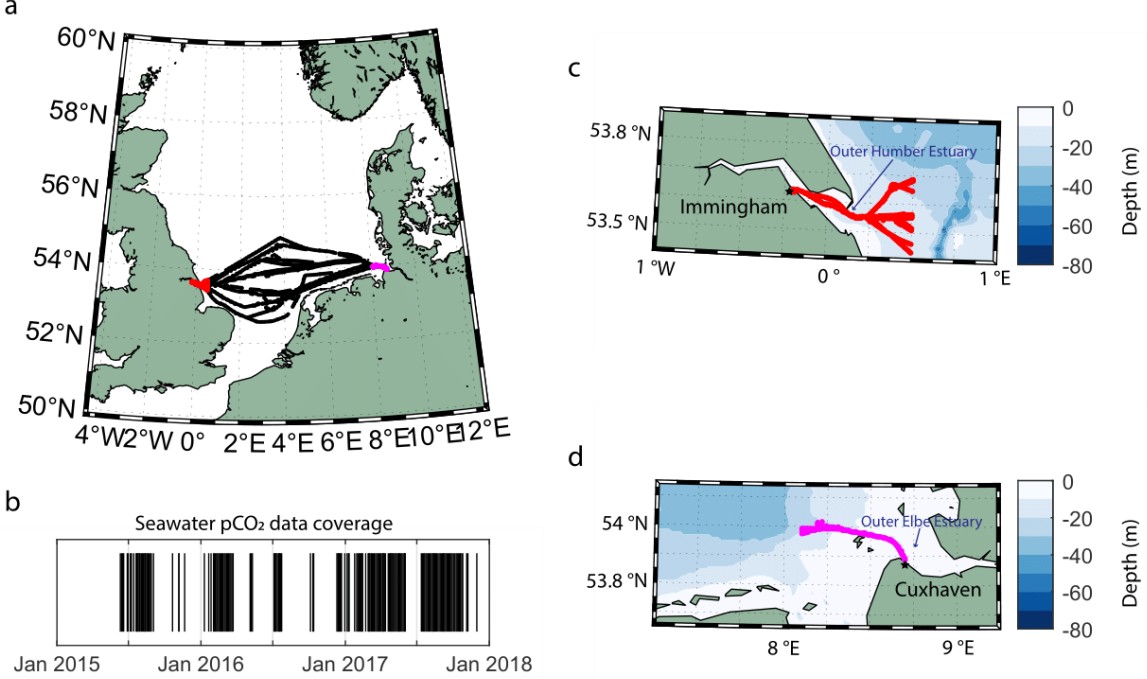


**Figure 1:** The SOO routes (a) and the temporal $p$CO$_2$ data coverage (b) of MS *Hafnia Seaways* travelling in the North Sea between 2015
and end of 2017. Black markers in panel b indicate the available measurements during this period. A zoom-in on the locations of the near-
shore measurements used in this study (red and magenta) and the NOAA ETOPO2 bathymetry near Immingham, UK (c) and Cuxhaven,
Germany (d) are also shown.






The Humber River catchment extends over 24,000 km$^2$ and its average river flow is England's largest at 250 m$^3$ s$^{-1}$ (Sanders et al., 1997). Its geology is Carboniferous limestone in the west and Permian and Triassic sandstone in the east, while the overlying Quaternary deposits are mostly clays (Jarvie et al., 1997b). The catchment area is a mix of industrialized, agricultural and urban areas, and the anthropogenically-influenced runoff affects the biogeochemical processes in the estuary

(Jarvie et al., 1997a). Between 70 and 80% of the catchment is arable land or grassland, and the agricultural practices cause the nitrate concentrations in the surface waters to frequently exceed the 50 mg L$^{-1}$ EU Nitrates Directive standard (Cave et al., 2003). The excess nutrients are carried to the estuary, which starts at Trent Falls, about 60 km inland from the coast, but estuarine primary production is strongly light-limited by high turbidity (Jickells et al., 2000). Therefore, most nutrients are likely transported offshore, rather than assimilated in the estuary. The estuary outflows of the Humber and other British east

coast rivers form the East Anglian plume, which influences primary production in the Southern North Sea further offshore of the immediate river and estuary outflows, reaching as far as the Southern Bight of the North Sea (Weston et al., 2004;Dyer and Moffat, 1998). The effluent from industrial sources increases the biological oxygen demand in the estuary, and therefore the Humber estuary oxygen concentrations are low (Cave et al., 2003). Tides in the Humber estuary are semi-diurnal and the tidal range of up to 7.2 m makes it a macro-tidal and well-mixed estuary.


The Elbe River catchment extends over 148,000 km$^2$, with an average river discharge of 730 m$^3$ s$^{-1}$, making it one of Europe's largest rivers (Schlarbaum et al., 2010). The source of the river is found in Czechia in the Giant Mountains, primarily made of granite. The Elbe then flows through sandstone mountains before crossing the flat fertile marshlands of north Germany. River water chemistry is strongly correlated with the watershed geology (Newton et al., 1987), so one would

expect the alkalinity in the Elbe to be lower than in the Humber River, which flows through limestone bedrock. However, Hartmann (2009) found that the carbonate abundance in silicate-dominated geology is also important. In addition, erosion rate, mean annual temperature, catchment area and soil regolith thickness, all can influence the river carbon chemistry (Lehmann et al., 2023). The anthropogenic pressure in the Elbe watershed is high, with most of the catchment being dominated by agricultural land use (Quiel et al., 2011). While water quality has improved since the 1980s (Amann et al.,

2012;Rewrie et al., 2023b;Dähnke et al., 2008), the measures have disproportionately reduced the phosphorus input, so the current nutrient load has an increased nitrogen to phosphorus ratio (Geerts et al., 2013). The Elbe Estuary extends from the Geesthacht Weir to the mouth of the estuary at Cuxhaven, Germany, a further 141 km downstream. The large port of Hamburg, situated in the upper estuary, also increases the anthropogenic pressure in the estuary due to regular dredging and industrial activities that facilitate the development of oxygen depleted zones (Geerts et al., 2017). Tides are semi-diurnal, and

with a tidal range of up to 4 m. Therefore, the Elbe Estuary falls between a meso- and macro-tidal estuary and the salinity profiles classify it as a partially-mixed to well-mixed estuary (Kerner, 2007).



## 2.2 FerryBox measurements

FerryBoxes are an integration of modular automated instruments that can be installed on SOOs to provide high-frequency
observations of sea surface waters (Petersen, 2014). A list of the instruments that were installed on the MS *Hafnia Seaways*
and which were used in this study is shown in Table 1. Quality controlled temperature, salinity and $p$CO$_2$ data obtained by
this SOO are now publicly available on the Pangaea repository (https://doi.org/10.1594/PANGAEA.930383) and have been
used in a previous study evaluating surface seawater $p$CO$_2$ trends (Macovei et al., 2021). All other data are currently
available in the European FerryBox Database (https://ferrydata.hereon.de).


**Table 1:** FerryBox-integrated instruments installed on MS *Hafnia Seaways*, measuring between 2015 and 2017 and used in this study.



| Parameter | Instrument | Manufacturer | Uncertainty |
|---|---|---|---|
| Seawater temperature and salinity | Citadel TS-N Thermosalinograph | Teledyne Technologies/Falmouth Scientific, United States | ±0.1 °C and ±0.02, respectively |
| Seawater $pCO_2$ | HydroC $CO_2$-FT membrane-based equilibration sensor | CONTROS Sensors, 4H-Jena Engineering GmbH, Germany | ±1% |
| Seawater pH | Ion Selective Field Effect Transistor (ISFET) | Endress+Hauser GmbH, Germany | ±2% |
| Total chlorophyll-*a* fluorescence-derived concentration and plankton species distribution | AlgaeOnlineAnalyser (AOA) | bbe Moldaenke GmbH, Germany | 0.01 µg L$^{-1}$ |
| Total chlorophyll-*a* fluorescence-derived concentration | ECO FLNTU Fluorometer and scattering meter | WET Labs, Sea-Bird Scientific, United States | 0.025 µg L$^{-1}$ |
| Turbidity | Turbimax W CUS31 Turbidity sensor | Endress Hauser GmbH, Germany | <5% |
| Chromophoric Dissolved Organic Matter (CDOM) fluorescence-derived concentration | microFlu Fluorometer | TriOS Mess- und Datentechnik GmbH, Germany | ±5% |
| Dissolved Oxygen | 4330F Optode | Aanderaa Instruments, Xylem Analytics, Germany | ±2 µM or ±1.5% |

The FerryBox system initiates the measuring phase automatically based on the GPS location after departure from port. This starts the flow of seawater and the recording of measurements. Since some instruments need time to reach optimal
functioning and since in this study we focus on data in proximity of ports, we have chosen to only use data from the ship's journeys arriving to port. Our investigation also revealed that the arrival time into ports is consistent, irrespective of the tidal stage or sea level (details in Supplementary Material Fig. S1). The times of high and low tide progress every day, so a long time series, like the ones used in this study, likely will cover all tidal stages with no bias.



## 2.3 Further processing of FerryBox data

The turbidity sensor was calibrated using discrete samples collected by the auto-sampler installed on board the vessel and measured in the laboratory between February 2016 and July 2017 ($R^2$=0.92, n=19). Measurements at the upper limit of detection or above were discarded.

The FerryBox was equipped with a bbe-Moldaenke AOA chlorophyll sensor, which provided valuable information about the relative distribution of plankton classes contributing to the total pigment signal (Wiltshire et al., 1998). The sensor can usually differentiate between diatoms, green algae, blue-green algae and cryptophytes. However, the total chlorophyll measurements by this instrument were overestimated. Furthermore, during the study period, we replaced the AOA sensor with a newly calibrated sensor in December 2015, February 2016 and March 2017. Instrument replacements were not done for the FerryBox-integrated chlorophyll WET Labs sensor. In March 2016, water samples were collected via the autosampler onboard the vessel and analyzed in the laboratory for total chlorophyll-*a* using high-performance liquid chromatography (HPLC). To achieve consistent and accurate results, we first corrected the AOA values to the WET Labs ones, and then used the relationship between the WET Labs sensor and the HPLC samples ($R^2$=0.94, n=4) to get the final results. The linear relationships between the AOA and WET Labs sensors were calculated for four periods in each estuary with coefficients of determination ranging from 0.52 to 0.95 (details in Fig. S2).

Between February 2016 and November 2017, the performance of the ISFET pH sensor was evaluated eleven times during maintenance visits by measuring buffer solutions with pH values of 7.0 and 9.0. Using the time regression of the difference between the measurements and the standards, we assessed that the instrument exhibited a drift of 0.00045 pH units per day, which we corrected for before reporting the final results.

We used the Matlab CO2SYS toolbox (van Heuven et al., 2011) with the Cai and Wang (1998) K1 and K2 dissociation constants and the Dickson et al. (1990) K $SO_4$ constant to calculate dissolved inorganic carbon (DIC) concentrations from $p$CO$_2$ and pH (converted to the NBS scale). The concentrations were converted to µmol L$^{-1}$ using the Gibbs Seawater Toolbox for Matlab (McDougall and Barker, 2011). An empirical relationship between CDOM fluorescence, expressed on the quinine sulfate (QSU) scale and dissolved organic carbon (DOC) concentration was used to calculate the latter. This relationship was based on the measurements of DOC concentrations and CDOM fluorescence during the research cruise HE407 with RV *Heincke*, which took place in August 2013 in the outer Elbe Estuary and German Bight. Water samples were analyzed in the laboratory and the following linear relationship ($R^2$=0.67, n=10) was found with results from the Trios microFlu CDOM fluorometer integrated on the FerryBox on-board: $DOC\ [\mu mol\ L^{-1}] = 10.75 \times CDOM\ [\mu g\ L^{-1}] + 162.06$. While we acknowledge the limitations of using an empirical relationship based on a single cruise, this relationship is close to other literature references discussed below, and the resulting DOC concentrations fall within the range of direct water sample



measurements made by the Flussgebietsgemeinschaft Elbe. We find that converting the proxy CDOM results into estuarine DOC concentrations is useful for understanding the partitioning of dissolved carbon matter between inorganic and organic
fractions.

**2.4 External data**

Sea level data at the Immingham Docks were obtained from the British Oceanographic Data Centre (https://www.bodc.ac.uk/data/hosted_data_systems/sea_level/uk_tide_gauge_network/processed/). Sea level data at the
Cuxhaven Pier were obtained from the German Federal Waterways and Shipping Administration (WSV), communicated by the German Federal Institute of Hydrology (BfG) (https://www.pegelonline.wsv.de/gast/stammdaten?pegelnr=5990020). The reporting frequency of the datasets were 15 minutes and 1 minute respectively. In order to extract the spring-neap tidal cycle, sea level data were processed by running a moving maximum and a moving minimum calculation with a 100-hour window. The highest (lowest) difference between the 100-hour smoothed maximum and minimum sea level occurs during spring
(neap) tides, with a recurrence interval approximately matching the literature value of 14.77 days (Kvale, 2006). The times of the spring and neap tides were identified with a peak search function on the smoothed tidal range. Data were categorized by assigning measurements taken within ±24 hours of the identified peaks to spring tide or neap tide periods, respectively.

We cross-checked our findings with fixed-point salinity data from the Cuxhaven observing station on the southern shore of
the Elbe Estuary, used by Rewrie et al. (in prep.). The time span of this dataset is 2020-2021, a few years later than the *Hafnia Seaways* data, but the comparison to a station with a high temporal resolution is valuable. We also use Cuxhaven station biogeochemical data from autumn 2022 as further assessment. To the best of our knowledge, there is no equivalent biogeochemical observing station in the Humber Estuary.

Finally, we used the Drift App of the CoastMap Geoportal (https://hcdc.hereon.de/drift-now/) under CC BY-NC 4.0 license to simulate water mass movement in our study areas. This application specifies drift trajectories using the Lagrangian transport program PELLETS-2D (Callies et al., 2011). We simulated 24-hour backward trajectories from selected locations in the two outer estuaries starting both at the high and low tides during spring and neap tide conditions, which we selected from the sea level data.

**2.5 Carbon flux calculations**

Atmospheric carbon dioxide measurements were obtained from the Mace Head observatory in Ireland (World Data Centre for Greenhouse Gases, 2020). These are reported as dry air mole fraction ($x$CO$_2$), expressed in ppm. We used barometric pressure, dew point temperature and 10 m wind speed from the ERA5 reanalysis product (Hersbach et al., 2018) selected for





the Humber Estuary region, as provided by the Copernicus Climate Data Store (https://cds.climate.copernicus.eu). When estimating sea-to-air carbon fluxes, we calculated average values of the years 2015-2017 for the required input terms. The atmospheric $xCO_2$ was converted to $pCO_2$ using the saturated water vapor pressure ($P_v$) formula from Alduchov and Eskridge (1996):

$$Pv = 610.94\exp\left[17.625T_{dp}/(243.04 + T_{dp})\right]^{-5}$$

$$pCO_2 = xCO_2(P_b - P_v)/1.01325$$

, where $T_{dp}$ is the dew point temperature and $P_b$ is the barometric pressure. The sea to air carbon dioxide flux requires knowledge of the gas transfer velocity ($k$), which depends on the square of the 10 m wind speed ($U_{10}$) and on the dimensionless Schmidt number (D). According to Wanninkhof (2014), we used:

$$k = 0.00251U_{10}^2\left(\frac{660}{D}\right)^{0.5}$$

The gas exchange, and the propagated uncertainty, was then calculated using the function *co2flux* available online (https://github.com/mvdh7/co2flux) (Humphreys et al., 2018):

$$F = k\alpha(pCO_2^{sw} - pCO_2^{air})$$

, where the difference between the seawater and atmospheric partial pressures is multiplied by the gas transfer velocity ($k$) and the solubility of carbon dioxide ($\alpha$), a function of sea surface temperature and salinity (Weiss, 1974).

## 3 Results


The frequency of the repeating ship journeys to each port is too low to resolve the semi-diurnal high and low tides that characterize the two estuaries, but it is high enough to provide data during each stage of a spring-neap tidal cycle. Our observations were showing cyclical variability, so we applied spectral analysis on our dataset using the fast Fourier transform method with the Matlab *fft* function. The signal with the highest power had a period of 14.5 days for the Humber

estuary and 14.1 days for the Elbe estuary, indicating that spring-neap cycling is the main mode of $pCO_2$ variability for SOO data in these regions (Fig. S3). We applied the same analysis to continuous (1 minute resolution) salinity data from the Cuxhaven observing station. In this case, the plot of the Fourier coefficients power against the period produced an overwhelmingly strong peak at approximately 12.4 hours period, indicating that the semi-diurnal tides dominate the variability at the resolution provided by the station. When filtering out the high frequency data with a 3.5-day moving

average, a large peak in the power spectral density at 14.5 days revealed the spring-neap-driven biogeochemical variability in the station data too (Fig. S4).



### 3.1 Tidally-controlled biogeochemistry in the Humber estuary and adjacent coast

The SOO observations reveal that biogeochemical parameters in the outer Humber Estuary vary according to the spring-neap

tidal cycle. We selected all available measurements (Fig. 1c) around six positions based on the longitude (±0.005° around every 0.1°) and further split and sorted them into spring or neap tide measurements (Fig. 2). We chose these locations along the estuarine gradient because they demonstrate the changes from port, through the outer estuary and to the coastal sea, with sufficient distance covered to demonstrate differences between spring and neap tide conditions at the land-sea interface. For other methods of selecting the data with similar results, see Fig. S5.






**Figure 2:** Box plots of salinity (a), dissolved oxygen saturation (b), seawater $p\text{CO}_2$ (c), pH (d), total chlorophyll concentration (e), blue-green-like chlorophyll concentration (f), turbidity (g) and CDOM (h) grouped in six regions in the outer Humber Estuary, comparing the neap (blue) and spring (red) tide measurements. The box plots display the median, interquartile range and outliers. When the notches of two box plots do not overlap, they have different medians at the 5% significance level. Spring tide turbidity values are above the upper limit of detection in the westernmost two box plots in subfigure (g), and are therefore reported as this maximum value, without variance.





The differences between the mean for spring tide and that for the neap tide were tested with a Welch's t-test at the 1% confidence level. During spring tide, the salinity was significantly lower than during neap tide at all locations (Fig. 2a). Both the spring and neap tide selected datasets captured various stages of the semi-diurnal tidal cycle. For example, at the most upstream location the spring tide salinity during low tides was $18.9 \pm 0.3$ and during high tides $22.6 \pm 0.7$, while during neap tides, the salinity was $22.6 \pm 2.4$ during low tides and $25.8 \pm 3.2$ during high tides. Stronger oxygen undersaturation was also

observed during spring tides (Fig. 2b). Seawater $p\text{CO}_2$ was significantly higher during spring tide at the four westernmost locations, with a maximum median value of 787 µatm (interquartile range 700 to 844 µatm) (Fig. 2c). At the same four locations, spring tide pH was significantly lower than neap tide pH, while surprisingly, spring tide pH was significantly higher than neap tide values at the offshore locations (Fig. 2d). The total chlorophyll-*a* (Fig. 2e) and that associated with the blue-green algal group (Fig. 2f) had high variance, but generally had higher values during spring tides. Turbidity was

significantly higher at spring tide than at neap tide at all chosen locations (Fig. 2g). CDOM had a reverse pattern to turbidity and $p\text{CO}_2$, with spring tide values significantly lower than neap tide values at the four westernmost locations (Fig. 2h). Most measured variables show a gradient between the estuary and the offshore regions. Salinity medians ranging from 21 to 31, were lower than the North Sea salinity (32 to 35), and the low salinity plume was observed up to 0.2 °E, or 7 km offshore from the estuary mouth. There was also a west-to-east gradient in the oxygen saturation measurements. Turbidity, total and

blue-green-like chlorophyll-*a*, and CDOM were all lower offshore than in the estuary. In fact, during spring tide, turbidity measurements exceeded the upper limit of the sensor (~74 Formazine Turbidity Units, FTU).

The results presented in Fig. 2 show that the outer Humber Estuary experiences two distinct states, depending on the spring-neap tidal cycle. The advantage of the repeating ship journeys is that the transition between these states and the cyclical

variability were observed. In order to show this, we chose a two-month period with relatively good data coverage and present the Humber estuary data as a Hovmöller plot (Fig. 3). We show the same variables as in Fig. 2 and we overlay a line plot of the tidal range to collocate the spring and neap tidal cycle with the observed changes in the biogeochemical parameters.








**Figure 3:** An example of fortnightly variability in biogeochemical parameters in the outer Humber Estuary matching the spring-neap tidal cycle. The time/longitude coordinates of the measurements are shown in black. The tidal range is shown with the blue line and its secondary axis is valid for all subfigures. Large data gaps are masked out with gray boxes. The variables shown are salinity (a), dissolved oxygen saturation (b), $pCO_2$ (c), pH (d), total chlorophyll (e), blue-green-like chlorophyll (f), turbidity (g) and CDOM (h). The trajectories of 24-hour backward simulations of the water mass movement starting at a selected location on the ship route at the high and low tide of the four spring tide and four neap tide events during the two-month period are shown in subfigure (i).

In the two-month period shown in Fig. 3, four spring tides and four neap tides occurred. The transition between the two tidal states was influenced by the tidal range by modulating the strength of the estuarine influence. For example, the offshore extent of the spring tide-driven conditions, as well as the maximum levels reached during the less-pronounced spring tide event (24 February 2016) were smaller for most variables presented here, except for salinity and $pCO_2$. Similar to the median conditions in 2015-2017 (Fig. 2), CDOM was higher during neap tide. Especially further upstream in the estuary, the physical and biogeochemical variability was large and correlated with the tidal cycles. From neap to spring tide, the seawater changed from a salinity of 15, oxygen undersaturation and high carbon dioxide oversaturation to a salinity higher than 25, oxygen oversaturation and carbon dioxide close to atmospheric balance. In the estuary region, turbidity was below 20 FTU during neap tides and above the detection limit of 74 FTU during spring tides.

The simulations of the water mass movement (Fig. 3i) show that the surface water parcel that reached our chosen location at peak high tide came from offshore and had travelled the route twice in the previous 24 hours, according to the typical diurnal tides encountered in this region. When the simulation was initiated at peak low tide, the water movement is in the opposite direction, with the origin of the water parcel being upstream of the chosen location. The spring tide simulations show that the water traveled a greater distance both inshore and offshore compared to the neap tide conditions.

## 3.2 Tidally-controlled biogeochemistry in the outer Elbe Estuary

In spite of the ship not entering the estuary channel, instead stopping at the mouth of the Elbe at Cuxhaven (Fig. 1d), the median of the salinity measurements at the easternmost location was low, at around 19 for both spring and neap tide (Fig. 4a). This region has been described as the outer Elbe Estuary (Rewrie et al., 2023b) and is known to feature rapid changes in $pCO_2$ with changing salinity (Brasse et al., 2002). The high interquartile range at this location makes differentiating spring and neap tide measurements more difficult. Salinity increased offshore off the estuary outflow, but unlike in the Humber, statistically significant lower salinity was observed during neap tides. All oxygen measurement medians (expressed as percentage saturation) were higher than 100%, except in the region closest to Cuxhaven, where a significant difference between spring and neap tide oxygen was observed, with neap tide oxygen significantly lower (Fig. 4b). Seawater $pCO_2$ was generally higher towards the estuary, and overall above the atmospheric level with medians ranging from 422 to 594 µatm.



Unlike in the Humber, in the outer Elbe Estuary there was no difference between the spring and neap tide in seawater $pCO_2$, except at two locations (8.39 °E and 8.69 °E) where the Welch's t-test showed that neap tide $pCO_2$ values were significantly higher than the spring tide $pCO_2$ values (Fig. 4c). Furthermore, in the outer Elbe Estuary, the seawater $pCO_2$ values were overall lower than in the Humber Estuary, and closer to atmospheric balance. There was no significant difference in spring-to-neap pH, with little variability, and medians between 8 and 8.05 depending on location (Fig. 4d). Similar to the Humber Estuary, the total and blue-green chlorophyll-*a* concentrations in the Elbe Estuary showed a high variance (Fig. 4e,f). Unlike in the Humber however, there was no statistical difference between spring and neap tide and the blue-green-like chlorophyll-*a* concentrations were not proportionally as high. Turbidity was generally below 20 FTU except for some outliers (Fig. 4g). Similar to the Humber Estuary, neap tide CDOM measurements (Fig. 4h) were higher than spring tide and the maximum measurements were recorded upstream, at lower salinities. We split the high-resolution fixed-point data from the Cuxhaven station in the same way as the ship data. There were statistically significant differences (Welch's t-test at the 1% confidence level) between spring and neap tide data for the biogeochemical variables we tested. Furthermore, investigating a two-month period in more detail clearly displays two-week cycle variability (see Supplementary Material Fig. S4). This study demonstrates that the spring-neap variability from a fixed-point station can also be observed from a regularly transiting ship, with the added benefit of determining the spatial extent of the estuarine influence on the coast.





**Figure 4:** Box plots of salinity (a), dissolved oxygen saturation (b), seawater $p$CO$_2$ (c), pH (d), total chlorophyll concentration (e), blue-green-like chlorophyll concentration (f), turbidity (g) and CDOM (h) grouped in six regions in the outer Elbe Estuary, comparing the neap (blue) and spring (red) tide measurements. The box plots display the median, interquartile range and outliers. When the notches of two box plots do not overlap, they have different medians at the 5% significance level.





Figure 5 presents data from the same period as in Fig. 3, but in the outer Elbe Estuary. The tidal range is again shown on a secondary axis, but this time the axis is reversed so that spring tide events are represented by the curve moving away from the river end on the right side of the diagram. The tidal range, usually between 3.25 and 4.25 m, was smaller than in the Humber Estuary. The typical conditions that coincided with the peak spring tide in the Humber Estuary – low salinity,

oxygen undersaturation, high $p$CO$_2$, high chlorophyll and high turbidity – here occurred with a delay after the spring tide (peaks in color changes in Fig. 5 happen after the apexes in the tidal range line). This time lag was between 4-5 days, when the tidal range was dropping and the system was close to entering a neap tide stage. Blue-green algae (Fig. 5f) were no longer such a major contributor to total chlorophyll concentration (Figure 5e) as in the Humber Estuary. Maximum turbidity (Fig. 5g) and CDOM (Fig. 5h) were respectively lower and higher than in the same period in the Humber Estuary (Fig. 3g

and 3h). The simulations of the water mass movement for the outer Elbe Estuary (Fig. 5i) show a similar behavior as found for the outer Humber Estuary in terms of the direction of movement at high versus low tide. Although having a smaller tidal range than the Humber, the surface water in the outer Elbe Estuary travels a greater distance within the semi-diurnal tidal variation compared to the Humber. However, unlike the Humber, there is no clear difference between the extent of the distance travelled by the surface water between spring and neap tide conditions. All trajectories roughly overlap with the

exception of two high spring tide and one low neap tide.







**Figure 5:** Variability in biogeochemical parameters in the outer Elbe Estuary during the same period as shown for the Humber Estuary in Fig. 3. The time/longitude coordinates of the measurements are shown in black. The tidal range is shown with the blue line and its secondary axis is valid for all subfigures. Note the reversal of the axis and change of tidal amplitude compared to Fig. 3. Large data gaps are masked out with gray boxes. The variables shown are salinity (a), dissolved oxygen saturation (b), $p$CO$_2$ (c), pH (d), total chlorophyll (e), blue-green-like chlorophyll (f), turbidity (g) and CDOM (h). The trajectories of 24-hour backward simulations of water mass movement starting at a selected location on the ship route at the high and low tide of the four spring tide and four neap tide events during the two-month period are shown in subfigure (i).

## 3.3 Comparing the Humber and Elbe outer estuaries

There are differences identified in the previous sections, so in Table 2, we provide a comparative summary of some correlations between the measured biogeochemical parameters in each outer estuary, as well as the relative phytoplankton algal class composition.

**Table 2:** Comparison of the Pearson correlation coefficients between seawater salinity and $p$CO$_2$ and other FerryBox-measured essential ocean variables (EOV), respectively. Data from the whole 3-year time series and whole study area are used, irrespective of tidal cycle. All coefficients are significant at the 0.01 level and the number of measurements for each correlation is indicated. Also shown is the relative contribution of plankton species in the two outer estuaries. The total chlorophyll-a concentration is split into these four constituents.

| | Humber | Elbe |
|---|---|---|
| | EOV-Salinity correlation coefficients | |
| Sea surface temperature | 0.13 (n=39908) | 0.25 (n=30202) |
| Seawater $p$CO$_2$ | −0.82 (n=44757) | −0.57 (n=30592) |
| Total chlorophyll | −0.69 (n=5140) | −0.18 (n=3648) |
| Turbidity | −0.76 (n=35959) | −0.56 (n=25164) |
| CDOM | −0.42 (n=35861) | −0.68 (n=25164) |
| Dissolved Oxygen | −0.16 (n=43158) | −0.39 (n=30728) |
| | EOV-$p$CO$_2$ correlation coefficients | |
| Sea surface temperature | 0.24 (n=40045) | −0.13 (n=30234) |
| Sea surface salinity | −0.82 (n=44757) | −0.57 (n=30592) |
| Total chlorophyll | 0.55 (n=5154) | −0.34 (n=3621) |
| Turbidity | 0.73 (n=36487) | 0.52 (n=23975) |
| CDOM | 0.39 (n=36403) | 0.52 (n=23975) |
| Dissolved oxygen | −0.27 (n=43294) | −0.05 (n=30516) |





|  | Phytoplankton class composition | |
| --- | --- | --- |
| Diatoms | 34.1% | 53.3% |
| Blue-green-like algae | 31.4% | 16.4% |
| Green algae | 31.0% | 27.6% |
| Cryptophytes | 3.2% | 2.7% |

The negative correlation coefficients between salinity and $pCO_2$, chlorophyll, turbidity, CDOM and oxygen suggest that all these parameters are higher in the low salinity endmember. Turbidity, CDOM and $pCO_2$ are all higher in the low-salinity endmember than in the shelf sea and therefore positively correlated between each other in this estuarine setting. The relative

contribution of blue-green-like algae was higher in the outer Humber Estuary than in the outer Elbe Estuary, where diatoms produced more than half of the observed phytoplankton chlorophyll.

**3.4 Spring-neap effects on sea-air carbon fluxes at the land-sea interface**

The seawater $pCO_2$ in the outer Humber Estuary was higher than the atmospheric level, indicating that this estuarine outflow

on the coast is a potential carbon source to the atmosphere. Moreover, during spring tides (Fig. 2c and 3c), the seawater $pCO_2$ was much higher (at the most upstream location increasing from a mean of $640 \pm 59$ µatm to $813 \pm 142$ µatm), leading to a stronger potential source of carbon dioxide to the atmosphere, which we calculate here. If this spring-neap cycle difference is not considered, assessments of the role of estuaries in regional carbon budgets might underestimate the importance of estuaries in the carbon cycle. We calculate average seawater $pCO_2$ at the same locations as the box plot

analysis (Fig. 2c) and use these averages and the average 2015-2017 atmospheric $pCO_2$ ($401 \pm 6.2$ µatm) to calculate carbon dioxide fluxes at each location and differentiate between the $CO_2$ flux during neap and spring tides. We use the climatological average of the wind speed in the Humber Estuary of 6.9 m s$^{-1}$. At 0.4 °E, where the riverine influence is limited, the average neap flux was $3.6 \pm 3.7$ mmol C m$^{-2}$ day$^{-1}$ and the average spring flux was $5.0 \pm 5.5$ mmol C m$^{-2}$ day$^{-1}$. At the most upstream location, the increase between neap and spring tide averages was around 74%, from $24.7 \pm 7.9$ to $43.0$

$\pm 17.1$ mmol C m$^{-2}$ day$^{-1}$, respectively. The uncertainties of the flux calculations are propagated using the uncertainty of the atmospheric $pCO_2$, the standard deviation of the mean of the seawater $pCO_2$ measurements at a certain location and tidal category, and the 20% uncertainty for the gas transfer velocity coefficient found by Wanninkhof (2014). Monthly averaged climatological wind speeds at our studied location range from 5.3 to 8.4 m s$^{-1}$, but we only use the annual average because our observations span multiple years and we do not differentiate by month. We defined an area the width of the river channel

upstream and the width of the observational coverage offshore and integrated the fluxes over the resulting boxes. Based on our definition of spring and neap tide periods, these occupy around 50 days in one year. Therefore, based on the 2015-2017 conditions, the outer Humber Estuary releases $2.1 \pm 1.2$ Gg C per year to the atmosphere during neap tide events, compared





to 3.4 ± 2.1 Gg C per year during spring tide events, changing the area-integrated sea-air flux of this estuary during spring tide by over 60%. This change does not refer to the other 265 days during transitional tidal periods.

## 4 Discussion

### 4.1 Drivers of biogeochemical variability

The spring-neap tidal cycles influence the strength of the carbon source to the atmosphere in outer estuaries, accounting for the largest variability in $p\text{CO}_2$ on timescales longer than semi-diurnally, based on Fourier Transform analysis (Fig. S3). In outer estuaries like the Humber Estuary, this results in a stronger area-integrated carbon source during spring tides by over 60% compared to neap tide conditions. In our observations, the periodicity of the biogeochemical changes, including salinity, turbidity, seawater $p\text{CO}_2$, dissolved oxygen saturation and chlorophyll, largely aligns with the spring-neap tidal cycle periodicity. The tidal influence on biological processes involved in the carbon cycle has been investigated before. In one study, changes in the timing of spring-neap cycle contributed 10-25% of the changes in the interannual modelled phytoplankton bloom timing (Sharples, 2008). Similarly aligning to the roughly two-week spring-neap variability was the life cycle of a copepod species that adapted to take advantage of the tidally-controlled phytoplankton pulses (Rogachev et al., 2001). Below, we attempt to identify the processes that force the outer Humber Estuary system to have the observed spring-neap $p\text{CO}_2$ tidal variability.

Estuaries are generally considered to be net heterotrophic environments (Smith and Hollibaugh, 1993). In contrast, shelf seas, such as the North Sea, are generally net autotrophic and their in situ primary production is related to the riverine nutrient loads (Kühn et al., 2010). Therefore, the outer estuary region is characterized by large gradients, not only in salinity, but also in other biogeochemical parameters, such as nutrient concentrations and pH (Kerimoglu et al., 2018). Furthermore, the effect of $p\text{CO}_2$ reduction when river water mixes with seawater with high buffering capacity has already been observed in other river systems (Mu et al., 2021). Changes in the metabolic state of the estuary can happen rapidly, and the spring-neap tidal cycle can modulate the extent of the metabolic switch. In our study, we observed this switch happening at different locations depending on the spring-neap tidal state (Fig. 3 and 5). The observed spring tide oxygen undersaturation and increase in seawater $p\text{CO}_2$ in the outer Humber Estuary are indicative of a net heterotrophic environment. While higher chlorophyll concentrations were also observed during spring tides, we postulate that this is not locally-produced biomass and does not lead to a decrease in seawater $p\text{CO}_2$ via autochthonous primary production. Water mass movement simulations suggest that the chlorophyll observed in the outer estuary during spring tides could be allochthonous and could be produced in the river or inner estuary. The similarity of the observed chlorophyll fluorescence to blue-green algae-like material also hints towards a more inland origin of the organic matter. The cells could be damaged in contact with salt water and the chlorophyll released to be observed by our instruments (Yang et al., 2020).





Estuaries are sites of intense organic matter transformation processes, often receiving the title of "biogeochemical reactors" (Canuel and Hardison, 2016;Voynova et al., 2019;Voynova et al., 2015;Dähnke et al., 2022). In estuaries, labile allochthonous organic matter is often remineralized within, and the Elbe Estuary is a good example site of this (Abril et al., 2002;Schulz et al., 2023). Over 70% of the organic carbon in the estuaries around the North Sea is converted to inorganic carbon and heterotrophic degradation is the largest contributor to estuaries acting as carbon sources to the atmosphere (Volta

et al., 2016). We found an increase in spring tide turbidity in the outer Humber Estuary of at least 18%. Since spring tides promote conditions that facilitate remineralization (Wang et al., 2019;Abril et al., 2004;Opdal et al., 2019;Grasso et al., 2018;Beck and Brumsack, 2012), the resulting net effect is an increase in spring tide coastal seawater $pCO_2$ because the estuary outflow extends further offshore.

**4.2 Comparison of organic matter transformations between the estuaries**

In the outer estuaries of Humber and Elbe, the negative correlation between CDOM and salinity and the positive correlation between CDOM and $pCO_2$ (Table 2) confirm our existing knowledge about the origin and estuarine distribution of this component of the riverine dissolved matter. For instance, CDOM was strongly negatively correlated to salinity and showed a conservative behavior indicative of a terrestrial source in a study of Norwegian rivers (Frigstad et al., 2020). Dissolved

organic carbon, for which CDOM can be used as a proxy, can feature a variety of behaviors in estuaries from conservative, to constant, to rapidly changing and this depends on flushing time and variations in the source of organic matter (García-Martín et al., 2021;Bowers and Brett, 2008). What is particularly noteworthy in our study is that higher CDOM concentrations were usually associated with the neap tides in both estuaries (Fig. 2h and 4h). This suggests that the fraction of organic carbon in the dissolved form in the outer estuary is larger during the neap tide periods than during spring tides. At

the same time, the inorganic carbon in the form of dissolved carbon dioxide also varies with the spring-neap cycle (Fig. 2c and 4c). This has an impact on the quantity and type of lateral carbon transport across the LOC. On a shorter timescale of variability, Chen et al. (2016) found examples of CDOM being higher during the ebb tide phase, when phytoplankton is not dispersed by the more dynamic conditions at high tide. Resuspension of bottom sediments at spring tide might also suppress the effect of terrestrial sources, usually the main input of CDOM to the estuary (Ferreira et al., 2014). High CDOM can also

come from the bacterial degradation of phytoplankton-derived detritus (Astoreca et al., 2009), and high turbidity can protect the detritus from photodegradation (Juhls et al., 2019), although the switch between tidal conditions might be too fast for this process to play a major role. We postulate that, in tidally-driven outer estuaries, cycling between the spring and neap tidal stages provides favorable conditions for dissolved organic material to be either brought to the surface from the benthos or transported to the outer estuary from further inshore during spring tides and subsequently transformed during neap tides into

a measurable form by our CDOM instrument.





**Figure 6:** Estimated dissolved organic carbon (a) and (b) and dissolved inorganic carbon (c) and (d) concentrations in the Humber and Elbe estuaries, respectively plotted against salinity and color-separated according to the spring-neap tidal cycle. Relationship between seawater $p$CO₂ and total chlorophyll in the outer estuaries of the Humber (e) and Elbe (f). The data points are colored by month.

The addition of dissolved organic matter in the outer estuary is supported by the non-linear distribution of CDOM-derived DOC against a theoretical linear salinity gradient when assuming conservative mixing (Fig. 6a,b), indicating a source of DOC within the estuary, at the land-sea interface. Although we do not capture salinities for the 100% freshwater term, some data points in the middle of the outer estuary lie above the line connecting the marine term to the lowest available salinity. The DOC enhancement at mid-salinities was pronounced during neap tides in the Humber; for example, at salinities of 22.5





± 1, the mean spring tide DOC concentration was 297 ± 106 µmol L$^{-1}$, while the mean neap tide DOC was 467 ± 86 µmol L$^{-1}$. There was no similar pattern in the Elbe, where at the same salinities as in the Humber example, the mean DOC concentrations were higher than in the Humber, but statistically indistinguishable between spring and neap tide. The ratio

between the forms of dissolved carbon favored the inorganic one. At a salinity of 20 ± 1, DOC was approximately 10% of the total dissolved carbon pool in the Humber and 18% in the Elbe. This is slightly lower than the 20% average in UK rivers (Jarvie et al., 2017). The calculated DOC concentrations for the outer Humber Estuary were higher than the Humber riverine end-member used by García-Martín et al. (2021), but in the range reported by Tipping et al. (1997) for the Humber and by Williamson et al. (2023) for the Trent tributary, and actually lower than the Humber value reported by Painter et al. (2018)

and Dai et al. (2012). The Elbe DOC concentrations were within the range of measurements conducted by the Flussgebietsgemeinschaft Elbe at the Cuxhaven station (FGG, River Basin Community; https://www.fgg-elbe.de/elbe-datenportal.html). The linear empirical DOC:CDOM relationship we used has a high intercept of 162 µmol L$^{-1}$. This means that there is DOC in the water even when the CDOM fluorescence readings are 0. This fraction of DOC does not contribute to the fluorescence. The fluorometer can only detect a specific fluorescent chemical group and nothing that does not

fluoresce. Furthermore, in a study in the coastal Arctic, another empirical relationship between DOC and CDOM quinine sulphate fluorescence found a similarly high intercept of 110 µmol L$^{-1}$ (Pugach et al., 2018). Mid-estuary non-conservative DOC enhancement, such as in our study, was also found by McKenna (2004), who attributed it to phytoplankton-derived autochthonous inputs. In our estuaries, the measured high chlorophyll concentrations correlate with low oxygen, so the primary production happens elsewhere, and the DOC we observe at neap tides is a result of transformations of allochthonous

organic matter. Alternatively, the high turbidity during spring tides could prevent our fluorescence-based instrument to detect the entire CDOM present in the surface waters.

The calculated DIC on the Elbe side was higher than other reported values in the Elbe Estuary or German Bight (Rewrie et al., 2023b;Reimer et al., 1999), but similar relationships with salinity in the outer Elbe Estuary were observed at certain

times during our period of observations, as shown in the Supplementary Material provided by Rewrie et al. (2023b). The DIC values reported here are however subject to the combined uncertainties of the measurements and the calculations. The $p$CO$_2$ – pH pair, which we used due to our data availability, has the highest calculation uncertainty, with a carbonate ion squared combined standard uncertainty of nearly 40 µmol kg$^{-1}$ (Orr et al., 2018). There were also DIC data in the Elbe outflow at salinities higher than 30 which did not follow the salinity relationship of the other data points (Fig. 6d). These

data were calculated using measurements from July-August 2017, a period when another study found elevated DIC concentrations in the Elbe Estuary (Rewrie et al., 2023a).

There are also differences between the two outer estuaries related to the dissolved oxygen and chlorophyll measurements. In an open-marine setting, oxygen and $p$CO$_2$ are strongly inversely correlated, a function of primary production and respiration

(DeGrandpre et al., 1997). In coastal regions however, there are distinct $p$CO$_2$-to-dissolved oxygen relationships due to





variability in Revelle factors and different sea-air equilibration times (Zhai et al., 2009). Both in the Humber and Elbe outer estuaries, $pCO_2$ and dissolved oxygen were inversely correlated, although in the Elbe, the correlation was weak. Combining this with the direct versus inverse correlations between $pCO_2$ and chlorophyll in the Humber and Elbe, respectively (Table 2), indicates that the two outer estuaries have a different metabolic behavior. Having high chlorophyll coinciding with low

oxygen and high $pCO_2$ suggests that the chlorophyll was not locally produced in the outer Humber Estuary. Instead, this location is the site where chlorophyll-containing organic matter likely produced upstream in the estuary was typically remineralized. During the summer and fall seasons however, in months when the minimum seawater temperature was higher than 12.5 °C, the relationship was closer to what is expected during conventional marine primary production (orange tone colors at $pCO_2$ levels below 550 µatm in Fig. 6e). When combining the whole-year data and including the very high $pCO_2$

brackish water-influenced measurements, the usual negative correlation changes. Due to the location of the ports, the ship was not entering the Elbe Estuary main channel as far upstream as the Humber Estuary (Fig. 1d and 1c, respectively), so in the outer Elbe Estuary, we were observing a more typically-marine behavior (Fig. 6f). The outer Humber Estuary is highly turbid and a location of organic matter degradation, while the organic matter in the Elbe is processed further upstream in the estuary, allowing the outer Elbe Estuary to take up some carbon through organic matter production, leading to the

differences in Fig. 6e,f.

Some studies found that higher chlorophyll concentrations in estuaries are associated with neap tides, when the water residence time is longer and conditions are calmer (Lucas et al., 1999;Trigueros and Orive, 2000;Domingues et al., 2010;Flores-Melo et al., 2018). Production from phytoplankton is limited by light and nutrient availability. In estuarine

settings, the high nutrient availability means the peak chlorophyll usually coincides with peak solar irradiance (Jakobsen and Markager, 2016). Over a longer seasonal term, this guides the onset of spring plankton blooms. On shorter time scales, the optimal conditions for primary production occur at the onset of stratification, as the estuary is shifting towards a neap tide stage, but also benefiting from the extra nutrients brought to the surface during the previous dynamic spring tide stage. This succession of events is what likely led to our outer Elbe Estuary observations (Fig. 5e). In contrast to the paradigm of high

chlorophyll at neap tides, and similar to what we observed in the Humber Estuary, the Tagus estuary in Portugal showed higher biomass during spring tides (Cereja et al., 2021). This was caused by resuspension of sediments and mixing of microphytobenthos into the water column, a phenomenon also described by Macintyre and Cullen (1996). The high chlorophyll concentrations we measured during spring tide in the Humber Estuary either have a similar origin, or alternatively, come from somewhere else and are therefore not locally-produced.


The dominant phytoplankton class were diatoms, while green algae made up around 30% of the total chlorophyll in both estuaries. However, the relative abundance of blue-green-like algae in the Humber was nearly double to the Elbe (31% versus 16%, respectively, Table 2). This was mainly driven by the increasing blue-green-like algae chlorophyll concentrations at spring tides (Fig. 2f and 3f). Blue-green algae, or cyanobacteria, are microscopic photosynthetic



prokaryotes, which are more common in freshwater (Iriarte and Purdie, 1993). Their occurrence in the North Sea is usually restricted to the Skagerrak-Norwegian Channel region (Brandsma et al., 2013). Although this study focuses on estuary-influenced regions, and cyanobacteria can actually dominate the plankton biomass in estuaries or be in phase with tidally-driven stratification events (Murrell and Lores, 2004;Eldridge and Sieracki, 1993), there are likely no relevant concentrations of cyanobacteria in this region of the North Sea. The instrument is possibly interpreting the fluorescence excitation from a

slightly different algal group as that coming from cyanobacteria. During a usual open-ocean spring bloom, the most efficient nutrient-utilizing plankton are the diatoms (Flores-Melo et al., 2018). What we could be observing in the outer Humber Estuary is a smaller scale version of this effect, where diatoms are outcompeting other algae during neap tides, and the latter are utilizing the spring tide niche for their development (Rocha et al., 2002). Alternatively, if the chlorophyll we observed at spring tides was not autochthonous, our observations could be influenced by how refractory the material is. Different

phytoplankton species release varying dissolved organic matter, and the organic matter from the blue-green-like algae is more resistant to degradation by microorganisms, therefore observable by our instruments (Osterholz et al., 2021).

## 4.3 Models versus observations

The carbon flux we calculated from the outer Humber Estuary to the atmosphere ranged between 3.6 mmol m$^{-2}$ day$^{-1}$

offshore at neap tide and 43.0 mmol m$^{-2}$ day$^{-1}$ near Immingham at spring tide. This places the outer estuary outgassing between estimates for the Southern North Sea at 2.1 mmol m$^{-2}$ day$^{-1}$ (Prowe et al., 2009) and Northwest European Shelf estuaries as a whole at 54-170 mmol m$^{-2}$ day$^{-1}$ (Kitidis et al., 2019). Excess dissolved inorganic carbon produced by respiration and remineralization makes estuaries carbon sources to the atmosphere. A study found that about 60% of the heterotrophic carbon in an estuary was lost by evasion to the atmosphere (Raymond et al., 2000). We show here that this

evasion happens along a gradient and varies according to the different stages in the tidal spring-neap cycle, and therefore the estuarine influence on nearshore seawater $p$CO$_2$ can extend further offshore than expected, depending on a reference point, with influence on regional flux calculations.

The importance of the LOC is now a more prominent topic in the literature and the knowledge gaps are identified as research

priorities (Legge et al., 2020). The areas adjacent to river mouths were necessary to be considered when closing the carbon budget of the Baltic Sea (Kuliński et al., 2011). We have evidence that high $p$CO$_2$ waters can be advected seaward due to river outflow and tidal exchanges (Reimer et al., 2017). The Southern Bight of the North Sea was a carbon sink over an annual integration with a flux of 2.27 mmol m$^{-2}$ day$^{-1}$, but including estuarine plumes in the calculation decreased the carbon uptake potential to 1.78 mmol m$^{-2}$ day$^{-1}$ (Schiettecatte et al., 2007). In spite of all this, some Earth System Models still lack

the implementation of estuarine systems because of the computational constrains of reproducing their variability (Regnier et al., 2013). We observed this effect in particular for the outer Humber Estuary region in a previous study (Macovei et al., 2022). A model that correctly replicated high chlorophyll observations in the central North Sea also replicated the associated





decrease in sea surface $pCO_2$. However, the same model underestimated seawater $pCO_2$ in the area of influence of the Humber Estuary, likely since it associated the high chlorophyll recorded during spring tides with carbon dioxide drawdown.

We show here that these variables are not necessarily negatively-correlated in outer estuaries (Fig. 6e, Table 2), where conflicting processes occur, and this can lead to incorrect $CO_2$ sea-air flux estimates. As all coastal environments, outer estuaries are also vulnerable to anthropogenically-driven climate change, and uncertainty regarding the direction of change remains. Resolving the competing and rapidly varying processes of the future river-influenced coastal seas first requires a thorough understanding of the present-day processes.

**5 Conclusion**

Estuaries are complex environments, with tides inducing large variability in biogeochemical parameters. Here, we used the high measurement frequency and multitude of sensors that FerryBoxes installed on a Ship-of-Opportunity allow and described the spring-neap tidal variability in two large outer estuaries draining into the North Sea and find that spring-neap variability plays an important role in modulating carbon fluxes at the land-ocean interface. In the macrotidal, well-mixed

Humber Estuary, seawater $pCO_2$ was up to 21% higher during the more turbid spring tides, under heterotrophic conditions, as shown by the widespread oxygen undersaturation. At the most upstream location in our study area in the outer Humber Estuary, the sea-to-air carbon dioxide flux increased from $24.7 \pm 7.9$ mmol C m$^{-2}$ day$^{-1}$ during neap tides to $43.0 \pm 17.1$ mmol C m$^{-2}$ day$^{-1}$ during spring tides. This means that the strength of carbon evasion from macrotidal, well-mixed estuaries could be underestimated if the fortnightly tidal cycle is not considered. Moreover, we showed that the estuarine outflow

influence stretched at least 7 km offshore and this is sometimes not correctly reproduced in models, as observed by Macovei et al. (2022) using another dataset. We described the competing processes forcing $pCO_2$ in the outer estuaries and showed how different biogeochemical parameters correlate. Spring tide conditions were associated with higher phytoplankton biomass, mainly driven by blue-green-like algae, which were remineralized in the outer estuary. Neap tide conditions were associated with higher CDOM, produced in the estuary, likely derived from allochthonous material. The dissolved carbon

pool in both outer estuaries studied here was dominated by the inorganic form, with DOC being less than 20% of the total. However, indications of non-conservative addition of DOC into the outer estuary were observed at mid-salinities, in particular in the Humber, where DOC concentrations during neap tides were 57% higher than during spring tide, altering the lateral flux ratio of dissolved organic to dissolved inorganic carbon across the land-ocean continuum depending on the spring-neap cycle. The conclusions presented describe well-mixed tidal outer estuaries, like the ones we studied here, and

this is an important first step in understanding the drivers of biogeochemical variability in classical estuaries before the complexity of global-scale land-sea interactions are assessed. Although the North Sea is one of the most studied marginal seas, its biogeochemistry is still not fully parameterised for integration into models, in particular in the freshwater influenced areas. With this study, we are following the call of the community by providing observations at the land-ocean interface and



we hope the results will be used to improve the performance of regional biogeochemical models, with ulterior upscaling into
Earth System Models and carbon budgeting.

**Code availability**

Code is available upon request to the corresponding author.

**Data availability**

Quality controlled temperature, salinity and $p$CO$_2$ data obtained by this SOO are now publicly available on the Pangaea repository (https://doi.org/10.1594/PANGAEA.930383). All data are also available in the European FerryBox Database (https://ferrydata.hereon.de). Data are also available upon request to the corresponding author.

**Author contribution**

**Vlad Macovei:** Conceptualization, Formal analysis, Writing – Original Draft, Visualization
**Louise Rewrie:** Resources, Writing – Review and Editing
**Rüdiger Röttgers:** Resources, Writing – Review and Editing
**Yoana Voynova:** Conceptualization, Writing – Review and Editing, Supervision, Funding acquisition.

**Competing interests**

The authors declare that they have no conflict of interest.

**Acknowledgements**

We would like to thank the DFDS Seaways and CLdN RoRo & Cobelfret Ferries companies, as well as the captains, officers and crews of *Hafnia Seaways* for facilitating our measurements on board their ship. Our sincere gratefulness goes to the FerryBox group engineers and scientists who installed the instruments and regularly serviced them. We thank Kerstin Heymann for the HPLC chlorophyll analysis. FerryBox activities were funded by the Helmholtz Association. Measurements
during the RV *Heincke* campaign in 2013 were conducted under the grant number AWI-HE407-00. Funding for this research was provided through the Helmholtz Association EU Partnering "SEA-ReCap – Research capacity building for healthy, productive and resilient seas" project and the Helmholtz Association "Changing Earth" program, as well as by project Horizon Europe project LandSeaLot, Grant Agreement number 101134575.



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
