# Peer review of "Spring-neap tidal cycles modulate the strength of the carbon source at the estuary-coast interface"

_EGUsphere, 2024_

## Author Response (AR1)

We thank the reviewers for their careful consideration of our article and the comments they provided, which helped to improve the standard of the manuscript. Please find our responses (in blue) below. In the updated manuscript, we also changed the DOC versus salinity plots to show the correct graphical representation. The original diagrams were constructed using the Pugach et al. (2018) formula and we omitted to change them in the combined figure that we included in the manuscript. The values in the text and the conclusions are not affected, and they do not need to be changed since they were correctly calculated with the appropriate DOC:CDOM empirical relationship for our study area. We have also used the opportunity of the revision to streamline and improve portions of the text.

**Reviewer 1:**

The manuscript investigates the influence of spring-neap tidal cycles on carbon dynamics in two North Sea estuarine systems (Humber and Elbe), presenting a comparative analysis supported by field observations. While the study benefits from a comprehensive dataset, several fundamental issues need to be addressed before the manuscript can be considered for publication. Below are the major concerns:

**Major Comments:**

**1. Data Visualization**

Consider implementing standardized cartographic conventions for improved clarity and scientific rigor.

- The cartographic elements in Figures 1, 3i, and 5i exhibit significant technical deficiencies in their presentation. The water mass trajectory plots lack scientific justification and supporting methodological documentation.

We have adjusted the mentioned maps to contain conventional cartographic elements such as a north arrow and a scale bar. Based on comments from Reviewer 2, we also changed to a cylindrical cartographic projection to maintain right angles between the coordinates. We ensured no overlapping axis labels remain. We also added elements to increase the clarity of the location of the zoom-ins, with boxes around the selected locations in the larger map, or the location of the data selection for the box plots. We have relocated the water mass trajectory plots to the Supplementary Material and added details on the methodology.

- The box plots require statistical significance indicators for between-group comparisons.

We added symbols to the box plots to indicate more clearly where statistical differences between the spring and neap tide groups exist.

- The tidal range representations (blue lines, Figures 3 & 5) need proper measurement annotations and methodological context.

We added the secondary axis for the blue lines for all subfigures, not just the first subfigure as in the initial manuscript, and annotated the spring tides. We included extra methodological context for how the tidal range was calculated, and why they are included in Figures 3 and 5.

**2. Analytical Depth and Data Integration**

The Discussion section currently provides only cursory analysis of parameter relationships. The study relies heavily on literature values rather than leveraging the original dataset, quantitative

assessment of individual contributors to carbon flux variations is inadequately addressed. Recommend incorporating detailed statistical analysis of the collected data to derive system-specific carbon flux contributions.

We removed some of the external references in the Discussion section and focused more on the opportunities provided by our dataset. In this updated version, we strengthen the quantitative assessment of carbon fluxes by also calculating lateral fluxes of dissolved carbon at the land-sea interface. We consider the results of this study a valuable contribution to the literature. We show spring-neap variability in many biogeochemical parameters, we show unexpected relationships between $pCO_2$ and chlorophyll, we show different partitioning of the dissolved carbon forms depending on spring-neap phase and all of these processes are extending offshore into the coastal sea. Due to the complexity of the system, we are limited in determining the individual contributions of each parameter to the carbon measurements, but we show the changes that occur in the integrated system. We advocate this information needs to be published and visible to the modelling community to improve future representations of this highly dynamic region.

**3. Methodological Documentation**

- The FerryBox measurement system requires more comprehensive documentation, suggest including:

    - Detailed technical diagrams of the FerryBox installation

We believe this is not a necessary addition to the manuscript. The list of sensors is given in Table 1 and the exact configuration of the flow cells inside the FerryBox does not influence the results. Instead, we direct the readers that want to learn more about the FerryBoxes to a journal publication (Petersen, 2014) and a white paper (Petersen and Colijn, 2017) whose goal is to describe these installations.

    - Systematic workflow documentation

We believe this is covered in the Methods section. We clarified the chlorophyll correction explanation, clarified the pH scales and choice of dissociation constants, and added a reference to a study that explains the $pCO_2$ data processing method.

    - Quality control procedures

We added details about the maintenance visits performed during the data collection period.

    - Calibration protocols

We added details about the calibrations applied to the raw data.

- The air-water CO2 flux calculation methodology should either be integrated into the results and discussion or removed entirely

While there is merit to keeping the methodology of this calculation in the Methods section, we can follow the Reviewer's suggestion and limit the description to just the references used to obtain the formulas used in the flux calculations. We are moving this shortened description in the section where the results are presented.

**4. Conceptual Framework and Literature Context**

The Introduction requires structural reorganization. Current emphasis on general estuarine carbon flux heterogeneity should be condensed. Recommend expanding the discussion of

spring-neap tidal influences on carbon dynamics. Strengthen the articulation of the study's novel contributions to the field.

We used the Reviewer's suggestions and shortened the Introduction paragraph containing the information on carbon flux heterogeneity and included more information on what is currently being researched on spring-neap tidal influence on biogeochemistry. We believe the final paragraph of the Introduction clearly states the goals of this study and we have strengthened the Conclusion section to more clearly round up the novelty of the study and the importance of the findings.

**Reviewer 2:**

**General comment :**

This manuscript reports a detailed analysis of FerryBox measurements (on route near surface measurements from ship of opportunity) acquired during 4 years (2015-2018) on a route in the North Sea between Cuxhaven (Germany) and Immingham (England). The FerreyBox allows to measure temperature and salinity, carbonate chemistry (pH and pCO2), oxygen and optical properties of seawater that allows to derive information on dissolved organic matter and phyto-plancton in seawater. This manuscript only focuses on the date acquired in the Elbe Estuary (Germany) and the Humber Estuary (England) which are both large temperate estuaries that are under the influence of tide.

The temporal resolution of the dataset allows mostly to infer conclusion on the influence of the spring-neap tidal cycle. The study shows this variability regime has an effect on the biogeochemical variability, in particular in inorganic and organic carbon concentrations, on both sites and over the gradients encountered from the land-sea interface to the adjacent coastal region. Air-sea fluxes of CO2 are estimated at the scale of the estuaries and the authors argued that an important increase of the estuarine carbon source to the atmosphere is associated with spring tide through changes in the metabolic state. They suggest that their study has strong implications for the correct estimation of carbon budgets in tidally-driven estuaries.

This study is a significant contribution because it shows some original results at the estuary-coast interface based. In my point of view, one strength of this study is to use FerryBox data collected when the ship is leaving or entering the port, whereas this often corresponds to the moment where ferryBox instruments are disconnected in order to protect the instrument from heavily loaded waters. The manuscript is well written and the results are convincing. The figures and tables are of good quality but could be improved. I would be glad to support the publication of this manuscript after some revisions (which I believe to be minor). My major concerns (detailed in the following sections) are related to some methodological points and the structure of the manuscript.

**Specific comments :**

In my opinion, one major point in this manuscript is the question of how adapted the temporal resolution of the sampling is compared to the variability of the studied process. I have the feeling that this needs to be more deeply discussed in the manuscript. I believe that the information from figures S1, S3 and S4 could be synthesized and presented as a figure in the main manuscript to discuss this question of temporal resolution / variability.

We use the Reviewer's suggestion and bring a synthesized figure from the Supplementary Material into the main manuscript and use it to demonstrate that the temporal resolution of our observations is appropriate to investigate the biogeochemical impact of spring-neap tidal variability.

The trajectories presented on figure 3 and 5 are not convincing. I would suggest to make a choice between two options : (1) Use the full potential of the trajectories in the manuscript by giving more details on the calculation, presenting them with a detailed figure and using the results to support the discussion or (2) keep their use in the actual form and just present them as supplementary material.

We have moved the water mass trajectory plots to the Supplementary Material, especially since they deal with variability at smaller temporal scales than the spring-neap cycles. Our intention was to help explain the differing salinity patterns between the spring and neap tidal stages and identify where the allochthonous chlorophyll is coming from.

The results section 3.1 and 3.2 are devoted to describe separately the results in both estuaries. I am wondering if a more synthetic description could be given in order to make section 3.1 and 3.2 a little bit shorter.

Sections 3.1 and 3.2 are simply describing the main result figures of the manuscript and these have a two-fold importance: demonstrating the capacity of the ship-based FerryBox observation system to resolve the spring-neap variability in the estuaries where measurements were taken, as well as showing that the spring-neap cycles are reflected in a large array of measured biogeochemical variables. Shortening these sections would restrict showing the multitude of biogeochemical variations occurring at the spring-neap scale. Moving the water mass simulations subfigures to the Supplementary Material also allowed us to shorten the length of text in Sections 3.1 and 3.2

The method section for inorganic carbonate chemistry measurements is lacking precision : What type of pH is measured $pH_T$ or pH-NBS ? Values of the dissociation constant are not conventional (Dickson et al., 2007) for open-sea oceanography and should be justified if adapted to estuaries? Why are the concentrations converted to µmol.L-1 whereas the dissociation constants are usually defined with mass amount ?

The ISFET sensor measures on the total scale, but our choice of appropriate low salinity dissociation constants in the settings of CO2SYS required a conversion to the NBS scale. The calculated DIC was indeed obtained in µmol/kilogram, but since the DOC calculations from the fluorescence-concentration empirical relationship provided the calculated DOC in units of µmol/litre, therefore we converted one to the other for ease of comparison. Furthermore, the DIC and DOC concentrations expressed as units of quantity over volume were used in the calculation of lateral fluxes by multiplying by the river discharge expressed in units of volume over time.

The calculation of air-sea $CO_2$ fluxes are also lacking precision : Why the Mac head station for atmospheric $CO_2$ ? Which wind data have been used and why using average values ? Using average values can lead to strong errors, because if periods of strong seawater $pCO_2$ can be specifically associated to stronger (or weaker) winds, this can lead to large differences in the air sea fluxes.

The Mace Head station in Ireland was chosen because it is a long-standing, well known station, commonly used in sea-air flux calculations of regional studies. The 2015-2017 average atmospheric dry mole fraction here was 404 ± 6 ppm, which we used in the calculations in the manuscript. Following the Reviewer's suggestion, we investigated the effect of choosing other nearby stations as data source. For example, at ICOS atmospheric research station Weybourne, on the east coast of the UK, the average atmospheric $xCO_2$ was 411 ± 9 ppm for the 2015-2017 period. At ICOS atmospheric research station Cabauw, in central Netherlands, the average atmospheric $xCO_2$ was 421 ± 18 ppm for the 2015-2017 period. These are proportionally small differences compared to the variability in the seawater term that we demonstrate in the manuscript. Below we show the effect that the choice of station would have on the calculated fluxes in mmol C $m^{-2}$ $day^{-1}$. The choice of station does not change the conclusions in the manuscript.

| Atmospheric Station / Tide | Most inshore location | Most offshore location |
| --- | --- | --- |
| Mace Head / Neap | 24.7 ± 7.9 | 3.6 ± 3.7 |
| Mace Head / Spring | 43.0 ± 17.1 | 5.0 ± 5.5 |
| Weybourne / Neap | 24.1 ± 7.8 | 3.0 ± 3.7 |
| Weybourne / Spring | 42.3 ± 17.1 | 4.3 ± 5.5 |
| Cabauw / Neap | 23.0 ± 7.9 | 2.0 ± 4.0 |
| Cabauw / Spring | 41.3 ± 17.1 | 3.4 ± 5.7 |

The wind speed data were extracted from the ERA5 reanalysis on the Copernicus data portal and selected for the 2015-2017 period at the grid cell closest to the Humber Estuary. We clarified this in the manuscript text. We use average values for the wind speed because the aim was to isolate the effect of the changes in seawater $pCO_2$ depending on the tidal stage. We are comparing the calculated fluxes using the average measured seawater $pCO_2$ during neap tides and spring tides, respectively, while keeping the other variables constant. While the Reviewer is correct in outlining the influence of wind speed on instantaneous fluxes, there is no indication of an association of higher seawater at spring tide with, for example, stronger wind speeds, therefore the calculations using the averages are appropriate for our aims.

**Technical corrections**

L182-185 : There is something not clear to me in the correction applied to AOA and Wetlab sensors for chlorophyll. This section could be clarified.

We shortened and clarified the text.

L351-352 : This last sentence of this section could be added to the conclusion rather than in the results section.

We moved the sentence to the Conclusions section.

Figure 1 :The longitudinal clusters used for figure 2 and 4 could be presented by areas on the maps of the estuaries in order to help the reader to visualize this 0.1° clusters.

We added markers on the maps to indicate the box plots data selection clusters.

Figure 1, 3 and 5 : The projection used for the maps conducts to figures that are not horizontal. Maybe another projection could be used?

We changed the projection used so that the lines of latitude and longitude are exactly horizontal and vertical, respectively.

---

## Referee Report (RR1)

The manuscript demonstrates substantial improvement post-revision, with robust data supporting its conclusions and clear relevance to regional carbon budgeting. While minor issues persist (e.g., statistical clarity, uncertainty quantification), these are addressable through revisions:

**Remaining Limitations:**

**Statistical Methodological Details:** The statistical approaches (e.g., Welch's t-test) and corrections for multiple comparisons (e.g., Bonferroni) are not explicitly described, potentially undermining result reliability. Clarification in the Methods or figure captions is recommended.

**Data Gaps and Continuity:** Missing data handling during Ferry Box maintenance (e.g., interpolation or exclusion) and systematic biases from sensor replacements (e.g., $pCO_2$ sensor changed four times) require further discussion.

**Uncertainty Quantification:** While atmospheric station variability was addressed, the covariance between wind speed temporal variability and $pCO_2$ (e.g., high $pCO_2$ during spring tides coinciding with low winds suppressing fluxes) remains unassessed. Sensitivity analyses or expanded error margin discussions are advised.

---

## Author Response (AR2)

As before, we thank the Reviewers for their comments, which have improved the manuscript. We are happy that the current state of the manuscript only requires minor revisions. We describe below (in blue) how we applied these suggested changes and we hope these comments are satisfactory for the Editor to approve the final manuscript for publication.

Reviewer #1:

The manuscript demonstrates substantial improvement post-revision, with robust data supporting its conclusions and clear relevance to regional carbon budgeting. While minor issues persist (e.g., statistical clarity, uncertainty quantification), these are addressable through revisions:

 Remaining Limitations:

Statistical Methodological Details: The statistical approaches (e.g., Welch's t-test) and corrections for multiple comparisons (e.g., Bonferroni) are not explicitly described, potentially undermining result reliability. Clarification in the Methods or figure captions is recommended.

We added the following text to the manuscript: "Statistical differences between groups are assessed using a Welch's t-test (Matlab function ttest2), with a rather strict significance level of 0.01 to avoid false positives. This statistical method tests the null hypothesis that two populations, with not-necessarily equal variances, have equal means, and is appropriate to compare our spring and neap tide groups."

Data Gaps and Continuity: Missing data handling during Ferry Box maintenance (e.g., interpolation or exclusion) and systematic biases from sensor replacements (e.g., $pCO_2$ sensor changed four times) require further discussion.

FerryBox maintenance does not interfere with data availability, since it is done only when the ship is in port. The analyses in the manuscript are performed with original quality-controlled measurements. While there are gaps in the dataset selected for this manuscript due to the ship not sailing or problems with the system, we show that we have enough consecutive repeating journeys to capture the spring-neap cycles and to characterize the biogeochemical differences between spring and neap tide conditions.

Sensor replacements are an integral part of long-term autonomous instrument deployments. They limit the potential influence of instrument drift on data quality. For most of the FerryBox parameters, water samples measured in the laboratory are used for sensor data correction. For the $pCO_2$ data that the reviewer mentioned, there is no reference material. Instead, we send the sensors to the manufacturer for pre- and post-calibration and we apply the corrections to the raw data in post-processing. This ensures a span-drift correction based on the runtime of the instrument. Finally, the data are checked for consistency around the times of sensor replacement to ensure no "jumps" in the data exist. These methods are described in Macovei et al. (2021a), where our data are compared to another ship's measurement. In Macovei et al. (2022), we find a very good match between our $pCO_2$ data and a Copernicus model in the central North Sea. The $pCO_2$ data are published in the PANGAEA repository (Macovei et al., 2021b).

The following sentences have been added/edited in the manuscript: "The pCO2 sensor was changed 4 times and appropriate data processing methods were applied, as described by Macovei et al. (2021b), to ensure a span-drift correction, and quality-controlled (Macovei et al., 2021c) to ensure no abrupt changes in the data occurred";  "All the statistical analyses are made

using original, quality-controlled data. There are sufficient data without gaps to capture consecutive spring and neap tide events and also to characterize the typical state of the system during the spring and neap tidal stages."

Uncertainty Quantification: While atmospheric station variability was addressed, the covariance between wind speed temporal variability and $pCO_2$ (e.g., high $pCO_2$ during spring tides coinciding with low winds suppressing fluxes) remains unassessed. Sensitivity analyses or expanded error margin discussions are advised.

There is no reason why the wind speed should be associated with the spring-neap tidal stage. The driving factors are different and disconnected. We checked this hypothesis by extracting daily averaged wind speed data in the Humber estuary from the ERA5 Reanalysis product for the first 6 months of the year 2017, when we also had a high data availability for $pCO_2$. We created a daily averaged dataset from our original tidal amplitude data to match the wind speed data (n=181). We found no correlation between these two datasets. A linear model fit produced a coefficient of determination of 0.003, and the correlation is insignificant (Pearson's p-value of 0.45). We added clarification text to the manuscript justifying that the choice of average wind speed is appropriate: "We found no correlation between the daily averaged wind speed data and the tidal amplitude (Pearson's p-value of 0.45, tested on 181 data points in the first six months of 2017). Since there was no association of the higher $pCO_2$ at spring tides with, for example, higher wind speeds, we therefore use the climatological average of the ERA5 wind speed in the Humber Estuary of 6.9 m s$^{-1}$. This isolates the investigated influence on sea-air carbon fluxes to tidally-driven seawater $pCO_2$, and not to wind speed."

Reviewer #2:

The authors have correctly revised the manuscript following my suggestions during the discussion. There are two minor points that could be further improved :

1 - The authors now mention that " We converted the ISFET pH measured on the total scale to the NBS scale before calculating," (L192 of the revised manuscript). The conversion between pHT and pHNBS is not so trivial to me. I recommend that the authors mention how this has been done.

The following sentence has been added to the manuscript text: "This conversion was performed using CO2SYS, with the associated values of temperature, salinity and $pCO_2$ for each pH value." The reference for CO2SYS is already given above in the text.

2 - In the response to my comments on the fact that average wind speed has been used to estimate air-sea CO2 fluxes, the authors have correctly justified there choice by mentioning "there is no indication of an association of higher seawater at spring tide with, for example, stronger wind speeds, therefore the calculations using the averages are appropriate for our aims." I would suggest to add this justification in the manuscript when mentioning that averaged wind speeds have been used.

We responded to Reviewer #1 above about a similar concern and statistically demonstrated the lack of correlation between wind speed and tidal amplitude. The following clarification text was added to the manuscript: "We found no correlation between the daily averaged wind speed data and the tidal amplitude (Pearson's p-value of 0.45, tested on 181 data points in the first six

months of 2017). Since there was no association of the higher $pCO_2$ at spring tides with, for example, higher wind speeds, we therefore use the climatological average of the ERA5 wind speed in the Humber Estuary of 6.9 m s$^{-1}$. This isolates the investigated influence on sea-air carbon fluxes to tidally-driven seawater $pCO_2$, and not to wind speed."

Aside from these minor changes, I would be happy to recommend this manuscript for publication.

Macovei, V. A., Voynova, Y. G., Becker, M., Triest, J., and Petersen, W.: Long-term intercomparison of two pCO2 instruments based on ship-of-opportunity measurements in a dynamic shelf sea environment, Limnology and Oceanography: Methods, 19, 37-50, 10.1002/lom3.10403, 2021a.
Macovei, V. A., Voynova, Y. G., Gehrung, M., and Petersen, W.: Ship-of-Opportunity, FerryBox-integrated, membrane-based sensor pCO2, temperature and salinity measurements in the surface North Sea since 2013. PANGAEA, 2021b.
Macovei, V. A., Callies, U., Calil, P. H. R., and Voynova, Y. G.: Mesoscale Advective and Biological Processes Alter Carbon Uptake Capacity in a Shelf Sea, Frontiers in Marine Science, 9, 2022.